# Direct evidence of substorm-related impulsive injections of electrons at Mercury

Sae Aizawa [1,2,3] ✉, Yuki Harada [4], Nicolas André[1], Yoshifumi Saito [2], Stas Barabash[5], Dominique Delcourt[6], Jean-André Sauvaud[1], Alain Barthe[1], Andréi Fedorov [1], Emmanuel Penou[1], Shoichiro Yokota [7], Wataru Miyake[8], Moa Persson[1,9], Quentin Nénon[1], Mathias Rojo[1], Yoshifumi Futaana[5], Kazushi Asamura[2], Manabu Shimoyama[5], Lina Z. Hadid[6], Dominique Fontaine [6], Bruno Katra[6], Markus Fraenz [10], Norbert Krupp[10], Shoya Matsuda[11] & Go Murakami[2]

Mercury's magnetosphere is known to involve fundamental processes releasing particles and energy like at Earth due to the solar wind interaction. The resulting cycle is however much faster and involves acceleration, transport, loss, and recycling of plasma. Direct experimental evidence for the roles of electrons during this cycle is however missing. Here we show that in-situ plasma observations obtained during BepiColombo's first Mercury flyby reveal a compressed magnetosphere hosts of quasi-periodic fluctuations, including the original observation of dynamic phenomena in the post-midnight, southern magnetosphere. The energy-time dispersed electron enhancements support the occurrence of substorm-related, multiple, impulsive injections of electrons that ultimately precipitate onto its surface and induce X-ray fluorescence. These observations reveal that electron injections and subsequent energy-dependent drift now observed throughout Solar System is a universal mechanism that generates aurorae despite the differences in structure and dynamics of the planetary magnetospheres.

The weak intrinsic magnetic field of Mercury interacts with the solar wind and creates a magnetosphere, which is similar to, but smaller than, that of Earth's. With a typical size about 5% that of Earth's, it has a typical standoff distance of the magnetopause of about 1.45 $R_M$ (Mercury radius, $R_M = 2440$ km)[1]. As a result, the magnetosphere experiences rapid rapid reconfigurations, following a cycle known as the Dungey cycle[2]. The Dungey cycle is a cyclinc behavior of magnetic reconnection between planetary magnetosphere and the solar wind. It involves fundamental processes observed at other magnetized planets like Earth, Jupiter, Saturn, and Uranus, including e.g., flux transfer events in the

dayside magnetosphere, plasmoids, and flux ropes in the magnetotail, and substorm-like activity[1]. During this cycle, plasmas are accelerated, transported, lost, or recycled throughout the magnetosphere. Substorm-related magnetic dipolarizations, a rapid reconfiguration of eroded magnetic field lines into a more dipolar state, have been observed in the tail of Mercury by Mariner 10[3] and MESSENGER (MErcury Surface, Space ENvironment, GEochemistry and Ranging[1]). These dipolarizations are associated with observations of in-situ electron flux enhancements at very high- to relativistic energy (>30 keV)[4–7] that may precipitate and induce X-ray fluorescence at the surface of Mercury[8–10],

[1]IRAP, CNRS-UPS-CNES, Toulouse, France. [2]Institute of Space and Astronautical Science, Japan Aerospace Exploration Agency, Sagamihara, Japan. [3]Department of Physics, University of Pisa, Pisa, Italy. [4]Department of Geophysics, Graduate School of Science, Kyoto University, Kyoto, Japan. [5]Swedish Institute of Space Physics, Kiruna SE 98192, Sweden. [6]Laboratoire de Physique des Plasmas (LPP), CNRS-Observatoire de Paris-Sorbonne Université-Université Paris Saclay-Ecole polytechnique-Institut Polytechnique de Paris, 91120 Palaiseau, France. [7]Department of Earth and Space Science, Graduate School of Science, Osaka University, Osaka, Japan. [8]Tokai University, Kanagawa, Japan. [9]Graduate School of Frontier Sciences, The University of Tokyo, Kashiwa, Japan. [10]Max Planck Institute for Solar System Research, Göttingen, Germany. [11]Kanazawa University, Kanazawa, Japan. ✉e-mail: sae.aizawa@irap.omp.eu

which results in producing secondary neutrals of the exosphere[11,12]. However, these observations were limited to the northern hemisphere of the magnetosphere because of the orbital coverage of those two missions, and no observations of electrons in the eV to keV energy range were available.

Here, we show the direct evidence that strongly supports the view that energetic electrons are accelerated in the near-tail region of Mercury's magnetosphere, drift rapidly toward the dawn sectors, and are subsequently injected onto closed magnetic field lines on the planetary nightside.

## Results and discussion
### Compressed magnetosphere of Mercury

The BepiColombo mission conducted the first of its six flybys of Mercury on 1 October 2021 and explored the southern hemisphere of the magnetosphere of Mercury. BepiColombo entered the magnetosphere through the nightside dusk magnetosheath and exited just forward of the dawn terminator (Fig. 1A). Among the seven sensors of the Magnetospheric Plasma Particle Experiment (MPPE)[13] onboard Mio (one of the two spacecraft of BepiColombo)[14], the two Mercury Electron Analyzer (MEA1 and MEA2), the Mercury Ion Analyzer (MIA) and the Energetic Neutrals Analyzer (ENA) simultaneously conducted plasma measurements inside the magnetosphere of Mercury (Fig. 1B–E), also see Supplementary Method). From these measurements, the crossing of various plasma boundaries of Mercury's magnetosphere along the BepiColombo trajectory could be identified, despite both a limited Field of View (FoV) due to the complex configuration of the spacecraft during cruise phase[15], and a reduced telemetry during the flyby that resulted in several data gaps (see Fig. 1). The inbound magnetopause (MP) crossing occurred at 23:08:30 UTC (Supplementary Fig. 1), whereas the outbound MP and bow shock (BS) crossing occurred at 23:41:00 UTC and 23:45:30 UTC, respectively[16], as also reported from other ion observations by the Search for Exospheric Refilling and Emitted Natural Abundances instrument onboard BepiColombo[17]. Due to a telemetry data gap, no inbound BS crossing was directly measured by MEA1, MEA2, or MIA. However, MIA and ENA observed different plasma populations before 22:45:24 UTC and after 22:49:53 UTC (Supplementary Fig. 1) which indicated that BepiColombo crossed the inbound BS during this time interval.

A comparison to the averaged BS and MP crossings by MESSENGER indicates that the magnetosphere was compressed, in particular, during the outbound leg of the flyby. A fit to the found BS and MP locations using the empirical models from previous studies (Fig. 2) yield extrapolated standoff BS and MP planetocentric distances of 1.67 and 1.22 $R_M$, respectively, which are smaller than the mean distances derived from MESSENGER observations of 1.90 and 1.45 $R_M$, respectively[18]. The inferred values correspond to the highest disturbance index (Dist = 100), which can be used in the KT17 magnetospheric field model[19]. As MIA has a limited FoV during

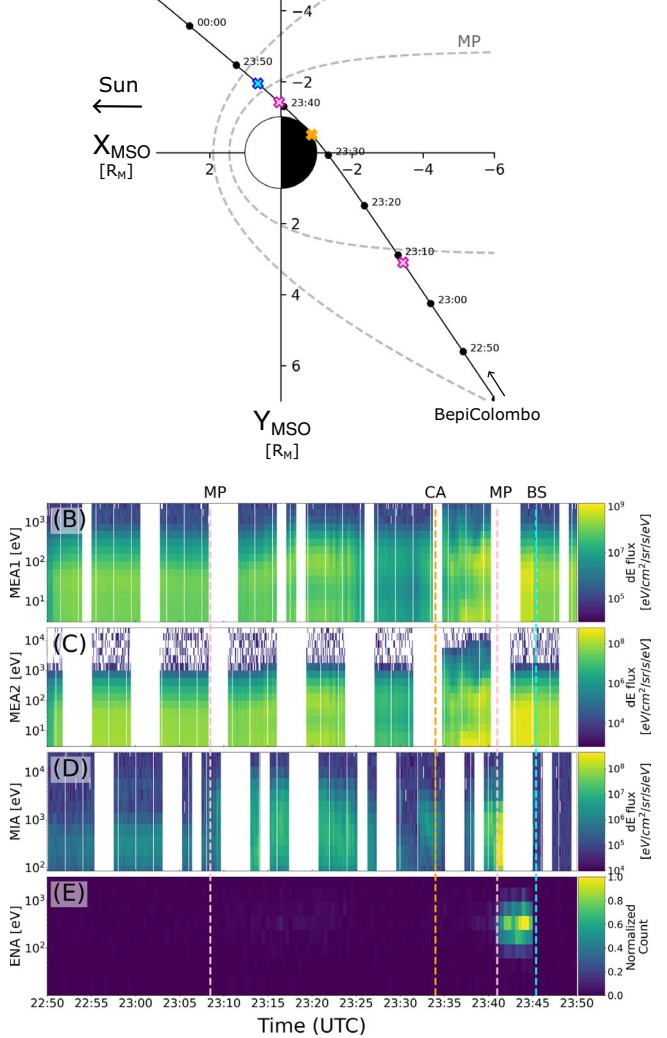

**Fig. 1 | Overview of Mercury Plasma Particle Experiments (MPPE) observations during the first Mercury flyby by BepiColombo on the 1 October 2021.**
**A** Trajectory of BepiColombo in the XY plane with time labels every 10 min in the MSO (Mercury-Solar-Orbital) coordinate system with average bowshock (BS) and magnetopause (MP) models (dashed lines)[18,39,40]. Pink, orange, and blue crosses correspond to identified both inbound and outbound magnetopause (MP), the closest approach (CA), and outbound bowshock (BS) crossings, respectively. Displayed times are in UTC [hh:mm]. **B–E** are energy-time spectrogram of electron omnidirectional differential energy flux from MEA1 (3 eV–3 keV), MEA2 (3 eV–26 keV), energy-time spectrogram of ion differential energy flux from MIA (25 eV–25 keV), and ions neutralized by the Magnetospheric Orbiter Sunshield and Interface Structure[13] before to be detected by ENA, respectively (see also Supplementary Fig. 1).

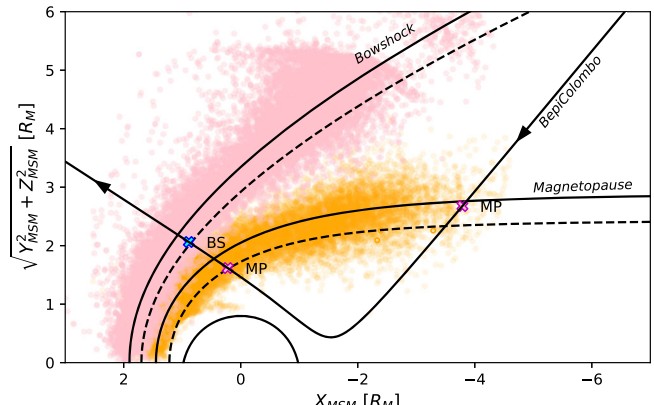

**Fig. 2 | Boundary locations detected by MPPE and compared to observations from previous space missions at Mercury, in the cylindrical Mercury-Solar-Magnetosphere (MSM) coordinate system.** Pink (orange) dots show all BS (MP) crossings identified from MESSENGER[41] and Mariner-10 observations. The solid black line with arrows shows the trajectory of BepiColombo during its 1 October 2021 flyby together with the outbound BS (blue cross) and inbound and outbound MP (pink crosses) crossings identified by MPPE. The two solid conic lines correspond to average BS and MP locations derived from MESSENGER observations, whereas the two dashed conic lines correspond to the fit to the outbound boundary locations identified by MPPE.

the cruise phase, it did not directly observe the solar wind. Thus, the solar wind dynamic pressure is estimated by considering both statistical values obtained by MESSENGER observations and a theoretical model of the compression of Mercury's magnetosphere that includes the effects of induction at Mercury's core[20]. This gives a solar wind dynamic pressure in the range of 28–60 nPa for the outbound leg of the flyby, which is more than two times higher than the most probable dynamic pressure of 10 nPa derived from MESSENGER observations[21].

## Low-frequency fluctuations on the dusk flank of Mercury's magnetosphere

Soon after BepiColombo entered the magnetosphere at dusk, and around the closest approach (CA), MEA and MIA detected two types of low-frequency fluctuations. The first type of fluctuations started at 23:21:30 UTC, lasted for 3.5 minutes, and is identified as Ultra Low Frequency (ULF) fluctuations with a period of 15–50 s (Fig. 3 and Supplementary Fig. 3A). These fluctuations likely correspond to either FLRs as observed by MESSENGER[22], or compressional Pc3 waves[23]. ULF

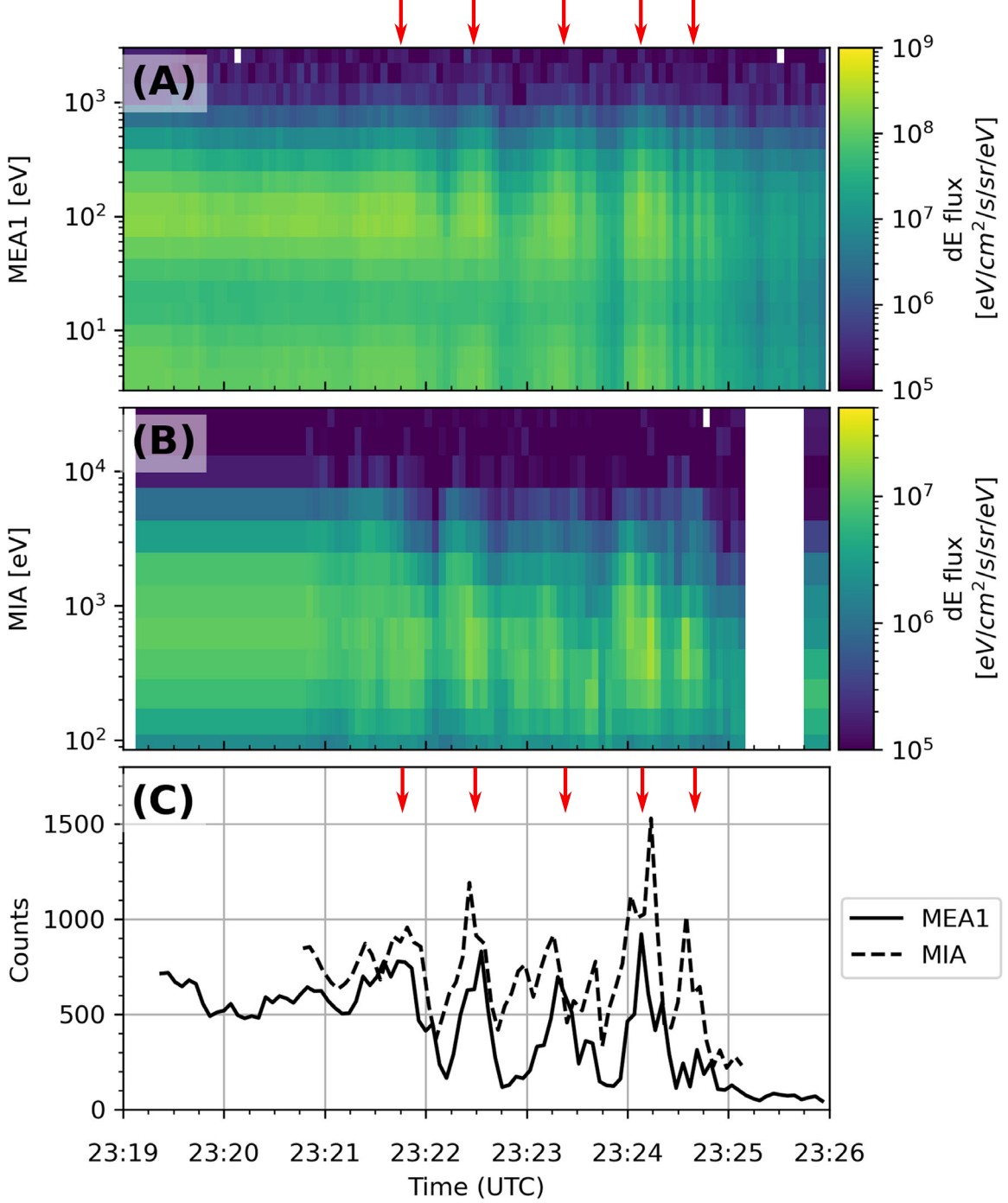

**Fig. 3 | Low-frequency fluctuations observed by both MEA and MIA at dusk.** **A** Energy-time spectrogram of electron differential energy flux from MEA1, **B** Energy-time spectrogram of ions differential energy flux from MIA, **C** Total electron and ion counts for both instruments. Red arrows indicate the identified electron fluctuations.

waves in the range of a few – 70 seconds encompassing our estimation were previously interpreted as Field Line Resonance (FLRs) events observed by Mariner-10[24] and MESSENGER[25]. The ULF fluctuations observed by BepiColombo were well inside the magnetosphere, in a region dominated by the intrinsic planetary magnetic field and tentatively identified as the low-latitude boundary layer (LLBL) from other ion sensor observations[17], and they were associated with plasma compression as indicated by MPPE. Since the boresights of the MPPE sensors were directed duskwards of Mercury at these times, these ULF fluctuations have also a component transverse to the magnetic field. FLRs in the magnetosphere are transverse waves without significant compressional component. At Mercury, similar events have been reported[25] and modeled[26] but they are with a mixture of compressional and transverse polarizations. Compressional waves were typically observed at low latitudes and transverse waves at high latitudes. This will be further investigated when calibrated magnetometer data will be available. In Earth's magnetosphere, ULF fluctuations are often associated with wave-particle interaction, which transfers energy from the inner to the outer regions of the magnetosphere, or from the solar wind to the magnetosphere. Interestingly, at Earth, ULF fluctuations observed in the LLBL were related to FLRs[27] induced either by a solar wind compression or Kelvin-Helmholtz vortices[28] at the magnetopause

which likely occur at Mercury[29], thus interactions with the solar wind are likely to be the main source of the observed ULF fluctuations.

## Energy-time dispersion of high-energy electrons

The second type of fluctuations started right after CA at 23:35 UTC, lasted for 5 minutes, and consisted of significant electron flux enhancements at energies between ~100 eV and a few keV observed by both MEA1 and MEA2 with a clear quasi-periodic fluctuation period of 30–40 s (Fig. 4 and Supplementary Fig. 3B). Note that the MIA data gaps prevent a discussion on if ions exhibit similar signatures or not. Figure 4 presents a detailed analysis of the corresponding six electron flux enhancements observed by both sensors. A remarkable feature is that 3 of the 6 electron flux enhancements (#1, #4, and #6) exhibit energy-time dispersion (Fig. 4D and Supplementary Fig. 4), a feature never observed before at Mercury.

The energy-time dispersion can be interpreted in terms of the gradient-curvature drift of electrons that are impulsively injected with broad energies and are subsequently trapped on closed magnetic field lines[30]. Since higher-energy electrons drift faster than lower-energy ones, the former arrives at an observation point first, follow by the latter. To illustrate the injected electron trajectories, test particle tracing in the KT17 model magnetic field with the

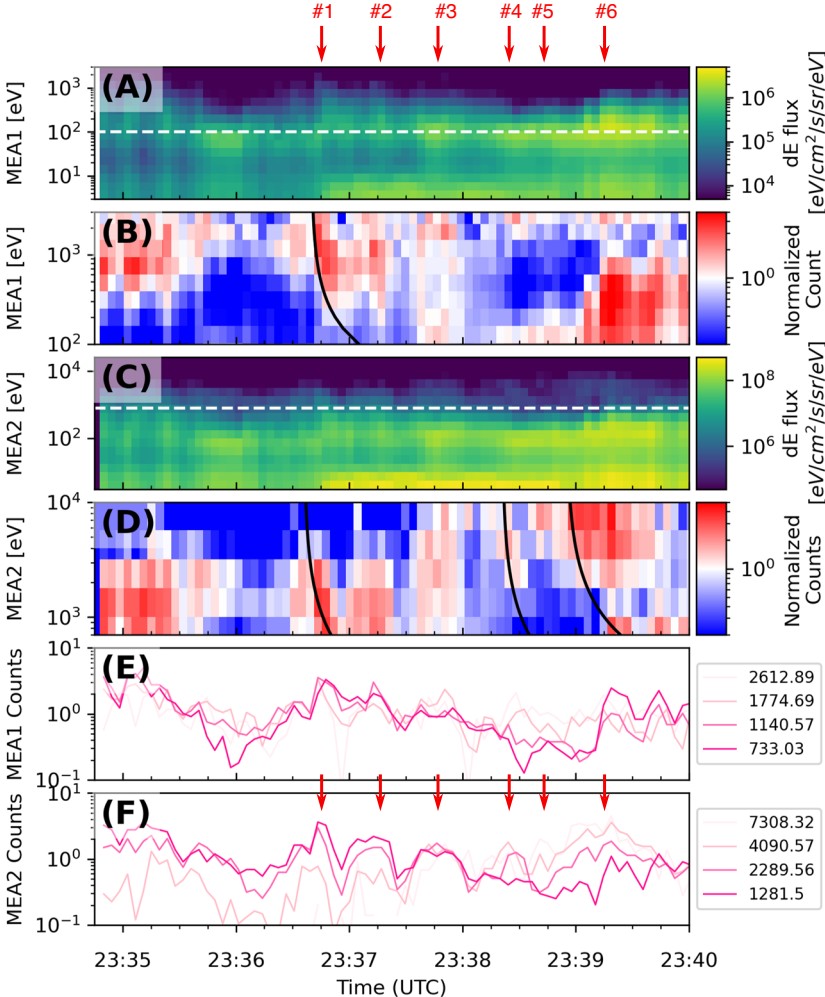

**Fig. 4 | Energy-dispersed electrons observed after CA. A** Energy-time spectrogram of electron differential energy flux from MEA1, **B** electron fluxes from MEA1 above 100 eV (dashed line in **B**-i) normalized by a 3 min window running average (23:36:37 – 23:39:31), **C** Energy-time spectrogram of electron differential energy flux from MEA2, **D** electron fluxes from MEA2 above 700 eV (dashed line in C) normalized by a 3 min window running average, **E** MEA1 electron counts, and **F** MEA2 electron counts at four selected energies. The red arrows indicate the electron enhancements reported. The black curves in **B** and **D** show fitted time ($\Delta t$)-energy($E$) dispersions with $\Delta t \propto 1/E$.

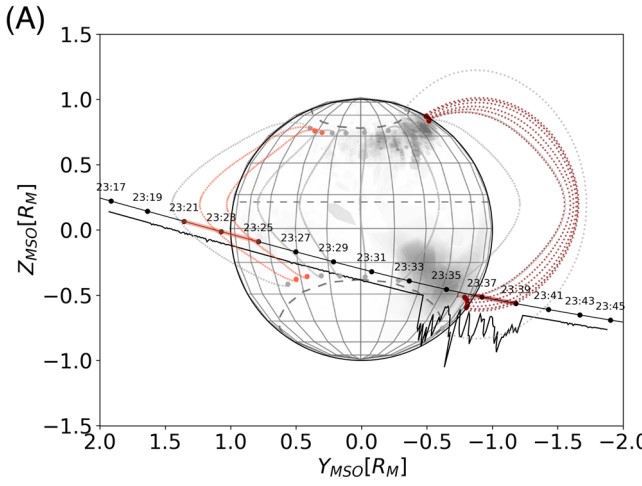

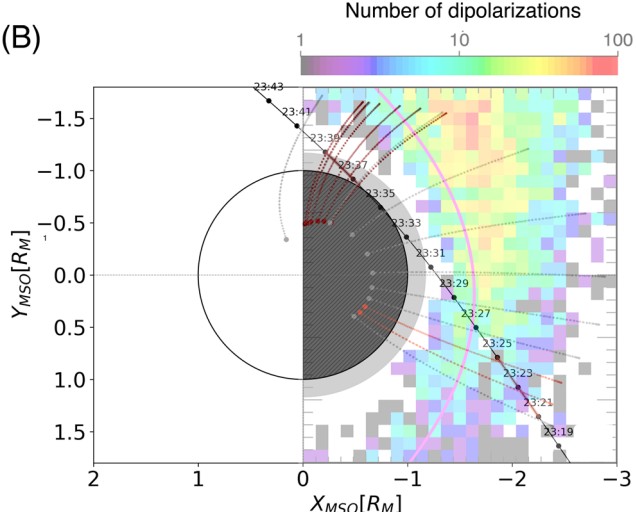

**Fig. 5 | Location of ULF fluctuation event and time-dispersed electron injections detected on closed magnetic field lines connected to the BepiColombo trajectory.** The magnetic field lines connected to the trajectory and the planet are obtained from the KT17 model (shown by gray dashed lines, see "Methods" section). Orange and dark lines represent the locations (solid, along the trajectory and dashed, the corresponding closed field lines) where the duskside ULF fluctuations and the energy-time dispersed electron injections have been observed, respectively. **A** Trajectory in the YZ plane (in MSO coordinates) projected onto a surface map of detected X-ray aurora by MESSENGER[10]. The times of the observations are marked on the trajectory of BepiColombo (solid black line), and the black line without marks represents the MEA2 count normalized by the event shown in Fig. 4. **B** Trajectory in the XY plane (in MSO coordinates) relating the electron injections detected by BepiColombo with the dipolarizations detected by MESSENGER[38]. 2D colors indicates the number of dipolarizations and the magenta line around X = −1 represents the plasma beta = 1.

highest disturbance index were conducted (19, see Method Field line tracing). The estimated injection regions by this method suggest that the source locations are relatively close to the magnetic field lines on which the energy-dispersed electrons were observed. Since MEA2 FoV may cover near-perpendicular pitch angles, the energy-dispersed electrons observed were likely trapped electrons on closed magnetic field lines in the post-midnight magnetosphere. The inferred close proximity to the injection source is consistent with the presence of the non-dispersive electron flux enhancements (#2, #3, and #5). The observed quasi-periodicity of ~30–40 s appears similar to the periodicity of high-energy (tens to hundreds of keV) electron bursts measured by Mariner 10 and MESSENGER,

that were attributed to drift echoes, which arise from a drifting electron cloud producing flux enhancements repeatedly at the drift period[6,30]. However, we find that the observed ~30–40 s periodicity of 1–10 keV electron flux enhancements are not consistent with the drift echoes, but is more likely to be caused by multiple electron injections during substorms. To have the period of 30–40 s, electrons drifting at an effective drift shell parameter of L = 2, must have energies of about 25 keV (see Methods Calculation of the drift echo Eqs.(1) and (2)), which is much higher than the observed energy of ~10 keV. Meanwhile, the periodicity of ~30–40 s agrees with the multiple onset time scale of 38 ± 3 s during a substorm at Mercury[7].

The ~1–10 keV electron flux enhancements were observed near the open-closed boundary in the post-midnight to dawn sector (Fig. 5A). These locations coincide well with those of a variety of signatures associated with substorms at Mercury, including energetic and suprathermal electron events[7], X-ray emission from the planetary surface induced by >3.5 keV precipitating electrons (Fig. 5A), dipolarization events (Fig. 5B), and fast plasma flows[31]. The observed periodic time-energy dispersion and locations of ~1–10 keV electron flux enhancements (Supplementary Fig. 4) and the fact that the occurrence rate of the diporalization fronts is bout two order of magnitude higher than the average under extreme solar wind conditions[21] jointly suggest that MEA observed substorm-related multiple injections of electrons on closed magnetic field lines that ultimately precipitate onto its surface. This evidence provides a critical link to complete our understanding of the dynamics of plasma populations inside the magnetosphere of Mercury[32]. The BepiColombo observations further reveal that electron magnetospheric injections and subsequent energy-dependent drift are remarkably similar to that at Earth despite the smaller scale of the magnetosphere of Mercury which is more sensitive to the external solar wind. Electron injections are therefore a universal mechanism in our Solar System that generates different types of aurora now observed at all magnetized planets except Neptune as well as above Martian crustal magnetic fields, despite the differences in structure and dynamics of their magnetospheres[33–35].

## Methods
### Field line tracing
The field line tracing was performed using the code stated in the code availability section. The model description can be found in Korth et al., 2017 and details and usage of the code can be find in https://github.com/mattkjames7/KT17. For each of the three dispersed electron signatures (#1, #4, and #6 in Fig. 4), we trace electrons with three different energies (2289, 4090, and 7308 eV) with variable pitch angles from the observed times and locations of the dispersion curves using the field line obtained by the field line tracing technique. The trajectories of the electrons are backwardly calculated and map the locations to the magnetic equator along the field lines at a given time. Injection time and region are estimated by finding a time at which the distance between the two equator-mapped locations becomes minimum (see Supplementary Fig. 5). The test particle trajectory in the given field is calculated using the full equation of motion.

### Calculation of the drift period
The drift period is roughly estimated to be

$$\tau_d \approx \frac{4\pi q M_0}{3 m_0 c^2 R_M} \frac{1}{L \gamma \beta^2} \tag{1}$$

Where

$$\gamma = 1 \Big/ \left(1 - \frac{v^2}{c^2}\right)^{\frac{1}{2}}$$
$$\beta = v/c \tag{2}$$

$v$ is the particle velocity, $q$ is the elementary charge, $M_O$ is the dipole magnetic moment of Mercury, $m_O$ is the electron mass, $c$ is the speed of light, $L$ is the drift shell parameter[36].

## Data availability

All data used to support the conclusions in this study are presented in the main paper or in the supplementary information. All the BepiColombo trajectory data and MPPE observation data, and field line tracing data presented here are available for download in the Zenodo database at https://doi.org/10.5281/zenodo.7926905 [37] and provided in the Source Data files. The Mio/MPPE observational data can be requested from the PI: Y.S. (saito@stp.isas.jaxa.jp) by describing the intent of the use of the data, as discussions with the respective PI is needed for analyzing the data, due to the complex configuration and operations of the BepiColombo mission during cruise phase. After the proprietary period of 12 months, the Bepi-Colombo mission data analyzed in this study will be available at the ESA-PSA archive https://archives.esac.esa.int/psa/#!Table%20View/BepiColombo=mission as soon as the data products are ready. The data obtained by MESSENGER and Mariner-10 mission that were used for Fig. 2 is available in Planetary Data System (https://pds-ppi.igpp.ucla.edu/index.jsp). DOIs for the Mariner-10 are https://doi.org/10.17189/1519732 and https://doi.org/10.17189/1519733, and the DOI for the MESSENGER is https://doi.org/10.17189/1522383. Source data are provided with this paper.

## Code availability

The code to perform the field line tracing is available at https://github.com/mattkjames7/KT17.

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

## Acknowledgements
S.A., N.A., D.D., J.-A. S., A.B., A.F., E.P., M.P., Q.N., M.R., L.Z.H., D.F., and B.K. acknowledge the support of Centre National d'Etudes Spatiales (CNES, France) to the BepiColombo mission. BepiColombo is a joint space mission between the European Space Agency (ESA) and the Japan Aerospace Exploration Agency (JAXA). MPPE is funded by JAXA, CNES, the Centre National de la Recherche Scientifique (CNRS, France), the Italian Space Agency (ASI), and the Swedish National Space Agency (SNSA). S.A. was funded by the French National Research Agency (ANR) for the TEMPETE (Temporal Evolution of Magnetized Planetary Environments during exTreme Events) project at the time of this work. The part of this work carried by S.A. is supported by JSPS KAKENHI number 22J01606. M.P. is funded by the European Union's Horizon 2020 program under grant agreement No 871149 for Europlanet 2024 RI. M.F. and N.K. are supported by the German Space Agency DLR under grant 50QW2101. The figures used in Fig. 5 are taken from Lindsay et al.[10] and Dewey et al.[38], and modified under the creative common license.

## Author contributions
S.A., Y.H., and N.A. conceived the study, analyzed the data, and wrote the initial draft of the manuscript. Y.H. modeled the injected electron trajectories. Y.S. is the Principal Investigator of the Mercury Plasma Particle Experiment (MPPE) consortium. G.M. is the project scientist of BepiColombo Mio for JAXA. All other co-authors, S.B., D.D., J.-A. S., A.B., A.F., E.P., S.Y., W.M., M.P., Q.N., M.R., Y.F., K.A., M.S., L.Z.H., D.F., B.K., M.F., N.K., and S.M. are Co-Principal Investigators, Co-Investigators, or collaborators on MPPE. All authors have read and provided feedback on the manuscript.

## Competing interests
The authors declare no competing interests.
