## [Peer Review File · Nature Communications]

REVIEWER COMMENTS

Reviewer #1 (Remarks to the Author):

Review of 'Direct evidence of substorm-related impulsive injections of electrons at Mercury'

By Aizawa, et al.

This paper presents evidence of substorm-related electron impulsive injections in Mercury's magnetosphere, based on in-situ plasma observations provided by BepiColombo during its first flyby around the planet on October 1, 2021. Making use of Mercury Electron Analyzer and Mercury Ion Analyzer observations, the authors conclude the Hermean magnetosphere was in a compressed state, as evidenced by the comparison between the observed magnetosphere boundaries compared to their expected location under nominal conditions. Moreover, the authors identified low-frequency waves on the dusk flank of Mercury's magnetosphere and characterized some of their properties. In particular, the second wave event observed right after BepiColombo's closest approach exhibits energy-time dispersion, a feature reported for the first time at Mercury.

In my opinion, the topic of the paper is definitively interesting for the space physics community, it is well-written and the conclusions are clear. However, I have a few comments and suggestions for the authors' consideration.

Line 54: Please add a reference describing the main properties and dimensions of Mercury's magnetosphere.

Line 55: Is the cycle the authors are making reference the Dungey's cycle? If so, please specify and add a reference.

Line 85: How do the authors identify the inbound magnetopause crossing location, despite the observed data gap in MEA1 and MEA2 (Figure 1B)? Is it only based on MIA data? I think the manuscript would benefit from a more in-depth description of Figure 1B.

Line 103: 28 to 60 nPa, instead of nT. Please correct.

Line 116: I think the interpretation in terms of field line resonances needs a justification. If these observations indicate plasma compression, how do the authors interpret this event as field-line resonance? I suggest putting these into context observations with previous studies, such as James et al 2019.

James, M. K., Imber, S. M., Yeoman, T. K., & Bunce, E. J. (2019). Field line resonance in the Hermean magnetosphere: Structure and implications for plasma distribution. *Journal of Geophysical Research: Space Physics*, 124, 211–228. <https://doi.org/10.1029/2018JA025920>

Line 173: Given the Hermean magnetosphere was in a compressed state, under an estimated solar wind dynamic pressure of 28 – 60 nPa, would the authors be able to provide a comment on how a nominal solar wind would affect the electron magnetospheric injection and energy-dependent drift processes in the Hermean magnetosphere?

Review of NCOMMS-23-00936-T:

Direct evidence of substorm-related impulsive injections of electrons at Mercury

S. Aizawa et al.

This paper presents initial results for electrons from the first flyby of Mercury from the BepiColombo spacecraft. Electron energy spectra are shown during the flyby which strongly indicate that there are impulsive injections of electrons into the inner magnetosphere of Mercury. The paper shows results from the MPPE instrument, in particular the Mercury Electron Analyzer (MEA), which for the first time at Mercury can directly observe electrons in the energy range less than 30 keV. This is particularly valuable since the MESSENGER spacecraft did not have an instrument capable of making direct measurements of electrons in the eV to ~ 30 keV range, although the XRS instrument on MESSENGER was able to detect the presence of 1-10 keV electrons indirectly [e.g., Ho et al. (2016), paper (3) in the Main references]. The results presented in this manuscript provide detailed electron energy spectrograms during the flyby, which indeed show strong evidence of electron injections and energy-time dispersion, strongly supporting the notion that ULF waves and magnetic substorm behavior occur at Mercury. Overall, the manuscript is very well written, nicely organized, and includes interesting and exciting new results. I highly recommend publication after addressing some minor questions and suggestions listed below.

Introduction Main Text line 63: please add a couple of references that discuss X ray fluorescence and other consequences of electron precipitation of Mercury. A reference to Starr et al. (2012) should be included since this was the first paper to discuss electron-induced X-ray fluorescence at Mercury. It is also suggested at this point to add another consequence of electron precipitation at Mercury, i.e., electron stimulated desorption (ESD), which can result in the emission of heavy ions from the surface into the magnetosphere (Schriver et al., 2011; McLain et al., 2011). Full references for these papers are given at the end of the review.

Line 103: Since pressure is being referred to, presumably it should be 28 \sim 60 nPa (not nT).

In the section starting on line 107, Low-frequency fluctuations on the dusk flank of Mercury's magnetosphere: Is there any magnetometer data available to make correlations with the particle data? Either way, if there is or isn't, it might be worth mentioning the status of magnetometer data as it relates to these flybys and what could be done in the future.

Figure 4, line 358: There is a reference to paper (8), but this should likely be referring to paper (7) by Lindsay et al.

Full references from above:

Starr, R.D., D. Schriver, L.R. Nittler, S.Z. Weider, P.K. Byrne, G.C. Ho, E.A. Rhodes, C.E. Schlemm, S.C. Solomon, and P.M. Trávníček (2012), MESSENGER detection of electron-induced x-ray fluorescence from Mercury's surface, *J. Geophys. Res.*, 117, E00L02, doi:10.1029/2012JE004118.

Schriver, D., P. Trávníček, M. Ashour-Abdalla, R.L. Richard, P. Hellinger, J.A. Slavin, B.J. Anderson, D.N. Baker, M. Benna, S.A. Boardsen, R.E. Gold, G.C. Ho, H. Korth, S.M. Krimigis, W.E. McClintock, J.L. McLain, T.M. Orlando, M.Sarantos, A.L. Sprague, and R.D. Starr (2011), Electron transport and precipitation at Mercury during the MESSENGER flybys: Implications for electron-stimulated desorption, *Planet. Space Sci.*, 59, 2026-2036, doi:10.1016/j.pss.2011.03.008.

McLain, J.L., A.L. Sprague, G.A. Grieves, D. Schriver, P. Trávníček, and T.M. Orlando (2011), Electron-stimulated desorption of silicates: A potential source for ions in Mercury's space environment, *J. Geophys. Res.*, 116, E03007, doi:10.1029/2010JE003714.

The authors acknowledge the constructive comments received from both reviewers. We have modified our manuscript accordingly.

Reviewer #1 (Remarks to the Author):

Review of ‘Direct evidence of substorm-related impulsive injections of electrons at Mercury’

By Aizawa, et al.

This paper presents evidence of substorm-related electron impulsive injections in Mercury’s magnetosphere, based on in-situ plasma observations provided by BepiColombo during its first flyby around the planet on October 1, 2021. Making use of Mercury Electron Analyzer and Mercury Ion Analyzer observations, the authors conclude the Hermean magnetosphere was in a compressed state, as evidenced by the comparison between the observed magnetosphere boundaries compared to their expected location under nominal conditions. Moreover, the authors identified low-frequency waves on the dusk flank of Mercury’s magnetosphere and characterized some of their properties. In particular, the second wave event observed right after BepiColombo’s closest approach exhibits energy-time dispersion, a feature reported for the first time at Mercury.

In my opinion, the topic of the paper is definitively interesting for the space physics community, it is well-written and the conclusions are clear. However, I have a few comments and suggestions for the authors’ consideration.

Line 54: Please add a reference describing the main properties and dimensions of Mercury’s magnetosphere.

→ Reference 1, “Mercury: The view after MESSENGER by Solomon et al., 2018” is now added on Line 54.

Line 55: Is the cycle the authors are making reference the Dungey’s cycle? If so, please specify and add a reference.

→ Yes, we referred to the Dungey cycle. We have made it clear on line 55. Our reference #2 on Line 58 that summarizes the discoveries made by MESSENGER already includes a reference to the Dungey cycle operating in the magnetosphere of Mercury.

Line 85: How do the authors identify the inbound magnetopause crossing location, despite the observed data gap in MEA1 and MEA2 (Figure 1B)? Is it only based on MIA data? I think the manuscript would benefit from a more in-depth description of Figure 1B.

→ The inbound magnetopause crossing is estimated by using both MIA and ENA observations as shown in our extended data fig 1. We have clarified it in the caption of the figure. In addition, we have now removed the original discussion on the inbound bow shock crossing on lines 93-94 because it cannot be identified accurately due to a data gap.

Line 103: 28 to 60 nPa, instead of nT. Please correct.

→ Thank you for pointing this out. Corrected.

Line 116: I think the interpretation in terms of field line resonances needs a justification. If

these observations indicate plasma compression, how do the authors interpret this event as field-line resonance? I suggest putting these into context observations with previous studies, such as James et al 2019.

James, M. K., Imber, S. M., Yeoman, T. K., & Bunce, E. J. (2019). Field line resonance in the Hermean magnetosphere: Structure and implications for plasma distribution. *Journal of Geophysical Research: Space Physics*, 124, 211–228. <https://doi.org/10.1029/2018JA025920>

→ Without magnetic field data, it is indeed not easy to identify the magnetospheric modes that we have observed. The first low-frequency oscillations reported in the manuscript were observed during the inbound leg of BepiColombo's trajectory when the magnetospheric boundaries were observed to be closed to their average locations, i.e. not compressed. The solar wind conditions have changed during the outbound leg of BepiColombo's trajectory as shown on Figure 2. Therefore, our interpretation in term of field line resonance for the first low-frequency oscillations reported cannot be ruled out. We have now added a reference to James et al., 2019 on line 118 to provide context observations with previous studies.

Line 173: Given the Hermean magnetosphere was in a compressed state, under an estimated solar wind dynamic pressure of 28 – 60 nPa, would the authors be able to provide a comment on how a nominal solar wind would affect the electron magnetospheric injection and energy-dependent drift processes in the Hermean magnetosphere?

→ The occurrence rate of the dipolarization fronts is about two orders of magnitude higher than the average under extreme solar wind conditions [Sun et al., 2016, included in reference number 22 in the manuscript]. Thus, we expect less electron injections under nominal solar wind conditions. We have now clarified it on lines 169-171.

Review of NCOMMS-23-00936-T:

Direct evidence of substorm-related impulsive injections of electrons at Mercury

S. Aizawa et al.

This paper presents initial results for electrons from the first flyby of Mercury from the BepiColombo spacecraft. Electron energy spectra are shown during the flyby which strongly indicate that there are impulsive injections of electrons into the inner magnetosphere of Mercury. The paper shows results from the MPPE instrument, in particular the Mercury Electron Analyzer (MEA), which for the first time at Mercury can directly observe electrons in the energy range less than 30 keV. This is particularly valuable since the MESSENGER spacecraft did not have an instrument capable of making direct measurements of electrons in the eV to ~ 30 keV range, although the XRS instrument on MESSENGER was able to detect the presence of 1-10 keV electrons indirectly [e.g., Ho et al. (2016), paper (3) in the Main references]. The results presented in this manuscript provide detailed electron energy spectrograms during the flyby, which indeed show strong evidence of electron injections and energy-time dispersion, strongly supporting the notion that ULF waves and magnetic substorm behavior occur at Mercury. Overall, the manuscript is very well written, nicely organized, and includes interesting and exciting new results. I highly recommend publication after addressing some minor questions and suggestions listed below.

Introduction Main Text line 63: please add a couple of references that discuss X ray fluorescence and other consequences of electron precipitation of Mercury. A reference to Starr et al. (2012) should be included since this was the first paper to discuss electron-induced X-ray fluorescence at Mercury. It is also suggested at this point to add another consequence of electron precipitation at Mercury, i.e., electron stimulated desorption (ESD), which can result in the emission of heavy ions from the surface into the magnetosphere (Schriver et al., 2011; McLain et al., 2011). Full references for these papers are given at the end of the review.

→ **Thank you for these additional references. We have now added the three references suggested on Lines 63-64, together with an explicit mention of the impact of electron precipitation on the emissions of heavy ions from the surface into the magnetosphere.**

Line 103: Since pressure is being referred to, presumably it should be 28 ~ 60 nPa (not nT). In the section starting on line 107, Low-frequency fluctuations on the dusk flank of Mercury's magnetosphere: Is there any magnetometer data available to make correlations with the particle data? Either way, if there is or isn't, it might be worth mentioning the status of magnetometer data as it relates to these flybys and what could be done in the future.

→ **Magnetometer data are not yet available. The instrument team is hardly working on removing disturbances and offsets on their measurements. Thus, we did not discuss the status of magnetometer data in the manuscript. However, a joint investigation between our plasma data and data from the magnetometer is expected in the future. We have clarified it on lines 121-122.**

Figure 4, line 358: There is a reference to paper (8), but this should likely be referring to paper (7) by Lindsay et al.

→ **Thank you for pointing this out. Yes, it should have been Lindsay et al., 2016 and we have now corrected it.**

Full references from above:

Starr, R.D., D. Schriver, L.R. Nittler, S.Z. Weider, P.K. Byrne, G.C. Ho, E.A. Rhodes, C.E. Schlemm, S.C. Solomon, and P.M. Trávníček (2012), MESSENGER detection of electron-induced x-ray fluorescence from Mercury's surface, *J. Geophys. Res.*, 117, E00L02, doi:10.1029/2012JE004118.

Schriver, D., P. Trávníček, M. Ashour-Abdalla, R.L. Richard, P. Hellinger, J.A. Slavin, B.J. Anderson, D.N. Baker, M. Benna, S.A. Boardsen, R.E. Gold, G.C. Ho, H. Korth, S.M. Krimigis, W.E. McClintock, J.L. McLain, T.M. Orlando, M. Sarantos, A.L. Sprague, and R.D. Starr (2011), Electron transport and precipitation at Mercury during the MESSENGER flybys: Implications for electron-stimulated desorption, *Planet. Space Sci.*, 59, 2026-2036, doi:10.1016/j.pss.2011.03.008.

McLain, J.L., A.L. Sprague, G.A. Grieves, D. Schriver, P. Trávníček, and T.M. Orlando (2011), Electron-stimulated desorption of silicates: A potential source for ions in Mercury's space environment, *J. Geophys. Res.*, 116, E03007, doi:10.1029/2010JE003714.

REVIEWER COMMENTS

Reviewer #1 (Remarks to the Author):

Review of 'Direct evidence of substorm-related impulsive injections of electrons at Mercury'

By Aizawa, et al. (NCOMMS-23-00936A)

I sincerely thank the authors for answering my previous comments. I have two additional comments for their consideration.

1) My previous comment about field line resonances having to be justified was meant to see if more information could be provided as the local plasma variability suggests compressibility. More specifically and based on Figure 3A, the plasma seems compressible between ~23:21:30 and ~23:25 UT. If this is correct, how do the authors reach the conclusion that these are possibly field line resonances under this condition? In addition, is the estimated wave period consistent with the results provided by James et al 2019 and Russel 1989?

Line 116: I think the interpretation in terms of field line resonances needs a justification. If these observations indicate plasma compression, how do the authors interpret this event as field-line resonance? I suggest putting these into context observations with previous studies, such as James et al 2019. James, M. K., Imber, S. M., Yeoman, T. K., & Bunce, E. J. (2019). Field line resonance in the Hermean magnetosphere: Structure and implications for plasma distribution. *Journal of Geophysical Research: Space Physics*, 124, 211–228. <https://doi.org/10.1029/2018JA025920>

-> Without magnetic field data, it is indeed not easy to identify the magnetospheric modes that we have observed. The first low-frequency oscillations reported in the manuscript were observed during the inbound leg of BepiColombo's trajectory when the magnetospheric boundaries were observed to be closed to their average locations, i.e. not compressed. The solar wind conditions have changed during the outbound leg of BepiColombo's trajectory as shown in Figure 2. Therefore, our interpretation in terms of field line resonance for the first low-frequency oscillations reported cannot be ruled out. We have now added a reference to James et al., 2019 on line 118 to provide context observations with previous studies.

Russell, C. T. (1989). Ulf waves in the mercury magnetosphere. *Geophysical Research Letters*, 16 (11), 1253-1256. doi: <https://doi.org/10.1029/GL016i011p01253>

2) In addition, reference (15), Orsini, S., et al. Inner southern magnetosphere observation of Mercury via SERENA 258 ion sensors in BepiColombo mission. *Nat Commun* 13, 7390 (2022). <https://doi.org/10.1038/s41467-022-34988-x> cited in this manuscript concludes that Bepi-Colombo crossed the latitude boundary layer (LLBL) between 23:10 UT and 23:25 UT, marking the transition between the Hermean magnetosheath and magnetosphere. I think the manuscript would benefit from a discussion between the signatures observed in Figures 3A and 3B and the observations provided by Bepi-Colombo PICAM and MIPA during the same time intervals.

Reviewer #2 (Remarks to the Author):

The authors have addressed all issues and comments raised in both referee reports satisfactorily and publication should proceed as soon as possible.

The authors sincerely thank the reviewers for reviewing our manuscript again and we acknowledge the comments received from both reviewers. We have modified our manuscript accordingly.

Reviewer #1 (Remarks to the Author):

Review of 'Direct evidence of substorm-related impulsive injections of electrons at Mercury' By Aizawa, et al. (NCOMMS-23-00936A)

I sincerely thank the authors for answering my previous comments. I have two additional comments for their consideration.

➔ **Thank you very much for reviewing our manuscript. Please find our answers to your comments below.**

1) My previous comment about field line resonances having to be justified was meant to see if more information could be provided as the local plasma variability suggests compressibility. More specifically and based on Figure 3A, the plasma seems compressible between ~23:21:30 and ~23:25 UT. If this is correct, how do the authors reach the conclusion that these are possibly field line resonances under this condition? In addition, is the estimated wave period consistent with the results provided by James et al 2019 and Russel 1989?

Line 116: I think the interpretation in terms of field line resonances needs a justification. If these observations indicate plasma compression, how do the authors interpret this event as field-line resonance? I suggest putting these into context observations with previous studies, such as James et al 2019. James, M. K., Imber, S. M., Yeoman, T. K., & Bunce, E. J. (2019). Field line resonance in the Hermean magnetosphere: Structure and implications for plasma distribution. *Journal of Geophysical Research: Space Physics*, 124, 211–228. <https://doi.org/10.1029/2018JA025920>

-> Without magnetic field data, it is indeed not easy to identify the magnetospheric modes that we have observed. The first low-frequency oscillations reported in the manuscript were observed during the inbound leg of BepiColombo's trajectory when the magnetospheric boundaries were observed to be closed to their average locations, i.e. not compressed. The solar wind conditions have changed during the outbound leg of BepiColombo's trajectory as shown in Figure 2. Therefore, our interpretation in terms of field line resonance for the first low-frequency oscillations reported cannot be ruled out. We have now added a reference to James et al., 2019 on line 118 to provide context observations with previous studies.

Russell, C. T. (1989). Ulf waves in the mercury magnetosphere. *Geophysical Research Letters*, 16 (11), 1253-1256. doi: <https://doi.org/10.1029/GL016i011p01253>

➔ **The ULF fluctuations were observed by BepiColombo well inside the magnetosphere, in a region dominated by the intrinsic planetary magnetic field and tentatively identified as the low-latitude boundary layer (LLBL) from other ion sensor observations (Orsini et al., 2022), and they were associated with plasma compression as indicated by MPPE. ULF waves in the range of a few – 70 seconds which our estimation is within were previously interpreted as Field Line**

Resonance (FLR) events observed by Mariner-10 (Russell, 1989) and MESSENGER (James et al., 2019).

FLR events in the Earth magnetosphere are transverse ULF waves without a significant compressional component, whereas similar events with a mixture of transverse and compressional waves have been reported (James et al., 2016) and modelled (Kim et al., 2015).

Interestingly, at Earth, ULF fluctuations observed in the LLBL were related to FLRs induced either by a solar wind compression or Kelvin-Helmholtz vortices at the magnetopause which likely occur at Mercury (Liljeblad and Karlsson, 2017). We have clarified this in the revised manuscript on lines 109-125, and included all references mentioned above.

2) In addition, reference (15), Orsini, S., et al. Inner southern magnetosphere observation of Mercury via SERENA 258 ion sensors in BepiColombo mission. Nat Commun 13, 7390 (2022). <https://doi.org/10.1038/s41467-022-34988-x> cited in this manuscript concludes that Bepi-Colombo crossed the latitude boundary layer (LLBL) between 23:10 UT and 23:25 UT, marking the transition between the Hermean magnetosheath and magnetosphere. I think the manuscript would benefit from a discussion between the signatures observed in Figures 3A and 3B and the observations provided by Bepi-Colombo PICAM and MIPA during the same time intervals.

→ LLBL is a region where both magnetosheath plasmas and magnetospheric plasmas exist. In the LLBL plasmas have higher temperature and density that gradually when moving closer to the inner magnetosphere, as shown in Lockwood and Hapgood (1997) for instance. In our opinion it is difficult to firmly conclude to the crossing of the LLBL from the SERENA observations since the PICAM data displayed in their paper are not given in physical parameters (only raw counts) and the raw data shown in their Figure 3 are displayed with a linear color scale showing limited variations (data not available unfortunately to assess quantitatively the variations, estimated by eyes to be around 20%) during the whole interval.

Based on the preliminary moments derived from our MIA observations that are restricted in field of view and that do not show the temperature and density profile (see Figure 1 below) expected for the LLBL (see Figure 2 below for a case study at Earth), we cannot firmly conclude that what we see here between 23:10 UT and 23:25 UT is the LLBL. We referred to Orsini et al. (2021) paper in the fairest possible manner by adding on line 112-113 of the revised manuscript ‘tentatively identified as the low-latitude boundary layer (LLBL) from other ion sensor observations (15)’. This will be investigated by combining all available plasma data together with calibrated magnetometer data when available.

Our Figure 3B only focus on newly-observed small-scale time-dispersed electron events in the Hermean magnetosphere during the time interval 23:35-23:40 UTC, whereas Orsini et al. (2021) report only large-scale ion observations. A careful examination of their Figure 5 shows that they have only three scans for PICAM ion observations during that time interval, which makes any comparison difficult in addition to the limitations already raised above. The MIPA ion data for that time interval are more time-resolved but,

unfortunately, they are also displayed as raw counts with very limited count rates.

Figure 1. Density-temperature plot from the preliminary moments derived from our MIA observations that are restricted in field of view. MSH, LLBL, and MSP stand for magnetosheath, Low-latitude boundary layer, and magnetosphere, respectively. The time interval for each region analysed is indicated directly in the figure.

Figure 2. Example of typical density-temperature profile (panel (c)) observed in the LLBL at Earth, adapted from the figure 3 in Nemecek et al., 2015, Analysis of

temperature versus density plots and their relation to the LLBL formation under southward and northward IMF orientations, JGR-Space Physics
(<https://doi.org/10.1002/2014JA020308>)

Reviewer #2 (Remarks to the Author):

The authors have addressed all issues and comments raised in both referee reports satisfactorily and publication should proceed as soon as possible.

→ Thank you very much for reviewing our manuscript.

REVIEWERS' COMMENTS

Reviewer #1 (Remarks to the Author):

I appreciate the authors' detailed responses to all my comments. I recommend the publication of this manuscript in Nature Communications.